# Catalytic Stereoselective Conversion of Biomass-Derived 4′-Methoxypropiophenone to *Trans*-Anethole with a Bifunctional and Recyclable Hf-Based Polymeric Nanocatalyst

**DOI:** 10.3390/polym13162808

**Published:** 2021-08-21

**Authors:** Yixuan Liu, Dandan Chen, Mingrui Li, Heng Zhang, Hu Li

**Affiliations:** State Key Laboratory Breeding Base of Green Pesticide & Agricultural Bioengineering, Key Laboratory of Green Pesticide & Agricultural Bioengineering, Ministry of Education, State-Local Joint Laboratory for Comprehensive Utilization of Biomass, Center for Research & Development of Fine Chemicals, Guizhou University, Guiyang 550025, China; gs.yixuanliu20@gzu.edu.cn (Y.L.); gs.ddchen20@gzu.edu.cn (D.C.); gs.mrli20@gzu.edu.cn (M.L.)

**Keywords:** biomass conversion, unconventional MOFs/polymeric materials, bifunctional catalysis, transfer hydrogenation, dehydration

## Abstract

Anethole (AN) is widely used as an odor cleaner in daily necessities, and can also be applied in the fields of food additives, drug synthesis, natural preservatives, and polymeric materials’ preparation. Considering environmental and economic benefits, the use of biomass raw materials with non-precious metal catalysts to prepare high-value fine chemicals is a very promising route. Here, we developed an acid-base bifunctional polymeric material (PhP-Hf (1:1.5)) composed of hafnium and phenylphosphonate in a molar ratio of 1:1.5 for catalytic conversion of biomass-derived 4′-methoxypropiophenone (4-MOPP) to AN via cascade Meerwein–Pondorf–Verley (MPV) reduction and dehydration reactions in a single pot. Compared with the traditional catalytic systems that use high-pressure hydrogen as a hydrogen donor, alcohol can be used as a safer and more convenient hydrogen source and solvent. Among the tested alcohols, 2-pentanol was found to be the best candidate in terms of pronounced selectivity. A high AN yield of 98.1% at 99.8% 4-MOPP conversion (TOF: 8.5 h^−1^) could be achieved over PhP-Hf (1:1.5) at 220 °C for 2 h. Further exploration of the reaction mechanism revealed that the acid and base sites of PhP-Hf (1:1.5) catalyst synergistically promote the MPV reduction step, while the Brønsted acid species significantly contribute to the subsequent dehydration step. In addition, the PhP-Hf polymeric nanocatalyst can be recycled at least five times, showing great potential in the catalytic conversion of biomass.

## 1. Introduction

With the industrial revolution, the human economy and technology have been developed rapidly [1], but it has also caused many negative issues like energy depletion and environmental pollution [2,3,4]. Therefore, the exploration of renewable energy has become an urgent matter [5,6,7]. According to data from the International Energy Agency (IEA), most of the power industry slows down in 2020 due to the response to the COVID-19 (Corona Virus Disease 2019) epidemic, but renewable energy still increased and accounted for about 90% of the total new power generation in 2020. According to the IEA’s forecast, by 2025, renewable energy may become the largest source of electricity all over the world. By then, renewable energy is expected to provide one-third of the world’s electricity, and its total capacity will be twice the current total power capacity in China. Therefore, as an important raw material for renewable and sustainable development, the irreplaceable role of biomass resources in the production of energy, materials, and chemical products has attracted widespread attention [8,9,10]. Compared with petroleum-based chemicals, the distinguishing characteristics of bio-based chemicals are their considerable water and oxygen content, thermal instability, and corrosiveness during the conversion process, which hinders the application and development of biomass resources [11]. Catalytic hydrogenation of oxygen-containing species or unsaturated bonds is considered to be a way to reduce the oxygen content of bio-based chemicals and increase their storage stability and heat of combustion [12]. Among them, the Meerwein–Ponndorf–Verley (MPV) reduction has attracted wide attention.

Anethole (AN) is widely used as a masking agent in commodities, such as soap, toothpaste, mouthwash, and cosmetics (Figure 1) [13]. It is also used as an additive and spice in food like candy, baked goods, chewing gum, cigarettes, beverages, and Chinese butter hotpot seasoning [14]. *Trans*-anethole (*trans*-AN) is also an important raw material for the synthesis of drugs, which has a certain therapeutic effect on leukopenia caused by chemotherapy or radiotherapy and other reasons [15]. In addition, AN plays a significant role in natural preservatives due to its remarkable ability to inhibit the growth of pathogens [16]. Further more, AN has also been reported to be used for preparing polymer materials by modifying its allyl and methoxy groups [17]. China’s imports and exports of unground star anise in 2020 are about 58 million dollars in total, as released by the General Administration of Customs of China. Anise, star anise, and fennel spices are used as sweeteners or for mouth fresheners, while only *trans*-AN is considered food grade. Therefore, a stereoselective catalyst is needed to maximize the proportion of *trans*-AN.

There are many kinds of industrial methods to obtain AN. For example, the natural preparation method is to cool anise oil to precipitate crystals, then distill and recrystallize with alcohol to obtain AN. Anise oil can also be rectified to collect the distillate at 230–234 °C or use the vacuum distillation method to collect 142 °C (5.60 kPa) or 110 °C (2.7 kPa) fractions to obtain AN. In addition, Ghazy et al. used the ultrasound-assisted method to prepare star anise extract nanoemulsion, with AN yield of 37%. However, the low content of essential oils in natural raw materials results in a large amount of waste that is inconsistent with the concept of sustainable development. Its yield, quality, and price fluctuate significantly due to climate constraints, so it is necessary to find a new and green approach to organic synthesis. In this regard, Lastra-Barreira et al. developed several ruthenium-based catalysts for the isomerization of estragole into AN [18]. It is worth mentioning that ruthenium, as a precious metal, does not have competitive economic costs with no visible industrial implications. As a promising alternative, Zr-MSU-3 was reported to be highly efficient for catalytic MPV reduction of 4′-methoxypropiophenone (4-MOPP). It can be derived from lignin to 1-(4-methyoxyphenyl)propan-1-ol (4-MOPE) that can be further dehydrated to AN (Figure 1). Nevertheless, it takes a relatively long reaction time of 23 h to achieve a satisfactory AN yield of 91.2% [19].

For the production of AN, almost all of the above methods have the disadvantages of low yield, complicated procedure, dangerous operation, and low catalyst reusability. Therefore, it is very meaningful and challenging to develop a more robust catalyst with a simple preparation process and high catalytic efficiency [20,21,22]. The use of suitable bifunctional catalytic materials simplifies the process of producing biofuels and high-value chemicals from lignocellulosic biomass [23,24,25]. The catalytic transfer hydrogenation (CTH) or MPV refers to the addition of hydrogen molecules to the unsaturated groups of organic compounds in the presence of a solid catalyst [26,27,28], which does not require external hydrogen or special catalytic hydrogenation equipment and is easy to operate, occupying a certain position in organic synthesis [29,30,31]. Therefore, the CTH reaction is recognized as a versatile method for upgrading bio-based carbonyl compounds [32,33,34,35].

Mesoporous metal-phosphonate hybrid polymeric materials can also be regarded as an unconventional MOF material, namely UMOF [36]. It has many advantages, including homogeneous composition, combined merits of inorganic units and organic groups, and considerable porosity [37,38,39]. Therefore, different ratios of phenylphosphonic acid (PhP) and HfCl_4_ were prepared by the solvothermal assembly to obtain hafnium phenylphosphonate (PhP-Hf) in this study. It can be used as an acid-base bifunctional mesoporous nanocatalyst for the high-efficiency production of AN.

## 2. Experimental Section

### 2.1. Materials

Phenylphosphonic acid (PhP, >98%), hafnium chloride (HfCl_4_, 99.6%), naphthalene (>99.7%) and *N*,*N*-dimethylformamide (DMF, >99.6%). Methanol (99.5%), ethanol (99.7%), 1-propanol (99.7%), 2-propanol (99.7%), 2-pentanol (98%), and 4′-methoxypropiophenone (>98%) were bought from Adamas Reagent Co. Ltd. (Shanghai, China). All other used reagents are of analytical grade.

### 2.2. Catalysts Preparation

PhP-Hf hybrids were prepared by a one-pot solvothermal method. In addition, 158.1 mg PhP (1 mmol) was added into a Teflon tube, followed by the addition of 60 mL DMF and stirring to completely dissolve before adding 453.4 mg HfCl_4_ (1.5 mmol). After stirring for 15 min to fully dissolve HfCl_4_, the tube was sealed and placed into an autoclave, followed by heating at 120 °C for 24 h. After cooling down to room temperature, the mixture was undertaken centrifugation at 5000 rpm, and washing successively with DMF, ethanol, and methanol. Then, the resulting precipitate was further dried at 80 °C. Upon grinding into powder, the target catalyst PhP-Hf (1:1.5) could be obtained. The value in parentheses represents the different molar ratios.

### 2.3. Catalytic Reaction

In a general synthesis procedure (Scheme 1), 4-MOPP, catalyst, solvent, and naphthalene as an internal standard were added into the reactor (WCGF-25 mL). In addition, the reaction time was set after the reactor was sealed in the oil bath. After the reaction stopped, the kettle was taken out of the oil bath and cooled to room temperature. Then, the liquid mixture was filtered before qualitative and quantitative analysis. All products were quantified by GC and liquid samples were identified by GC-MS. 

The conversion rate of 4-MOPP and the yield of trans-AN were calculated using the below formulae (deviation: ≤3.5%):(1)$$\mathrm{Yield}=\frac{\left(\mathrm{Moles}\;\mathrm{of}\;\mathrm{AN}\;\mathrm{or}\;\mathrm{ether}\;\mathrm{formed}\right)}{\left(\mathrm{Moles}\;\mathrm{of}\;4-\mathrm{MOPP}\;\mathrm{used}\right)}\times 100\%$$
(2)$$\mathrm{Conversion}=\frac{\left(\mathrm{Moles}\;\mathrm{of}\;4-\mathrm{MOPP}\;\mathrm{converted}\right)}{\left(\mathrm{Moles}\;\mathrm{of}\;4-\mathrm{MOPP}\;\mathrm{used}\right)}\times 100\%$$
(3)$$\mathrm{TOF}=\frac{\left(\mathrm{Mole}\;\mathrm{of}\;\mathrm{converted}\;4-\mathrm{MOPP}\right)}{\left[\left(\mathrm{Mole}\;\mathrm{of}\;\mathrm{acid}-\mathrm{base}\;\mathrm{sites}\;\mathrm{per}\;\mathrm{gram}\right)\times \left(\mathrm{catalyst}\;\mathrm{weight}\right)\times \left(\mathrm{time}\right)\right]}$$

### 2.4. Characterization Methods

Transmission electron microscopy [(HR)-TEM; JEM-1200EX] was applied to attain magnified sample images. The aberration-corrected FEI Tecnai TEM was used to obtain high-angle annular dark-field (STEM-HAADF) mappings. FT-IR (Fourier Transform Infrared Spectrometer) spectra were obtained by PerkinElmer 1710 (Thermo Fisher Scientific, Waltham, MA, USA). The XRD (X-ray diffraction) pattern was measured on D/max-TTR III (Rigaku, Akishima, Japan). Brunauer–Emmett–Teller (BET) was determined on a Micromeritics ASAP 2020 system (Micromeritics, Norcross, GA, USA). Thermogravimetric (TG) analysis was calculated using NETZSCHSTA 429 (NETZSCH, Selb, German). The TPD–CO_2_ and TPD–NH_3_ analyses were texted by the Micromeritics AutoChem 2920 chemisorption analyzer (Micromeritics, Norcross, GA, USA). The instrument used for X-ray photoelectron spectroscopy (XPS) is the Physical Electronics Quantum 2000 Scanning ESCA Microprobe (Physical Electronics Inc., Chanhassen, MN, USA). The model of the inductively coupled plasma optical emission spectrometer (ICP-OES) is PerkinElmer Optima 5300 DV (LabX, Midland, Canada).

## 3. Results and Discussion

### 3.1. Catalyst Characterization

FT-IR (Figure 2a) was examined to determine the structural functionalities of the catalyst. Bands at around 560 and 760 cm^−1^ represent Hf-O bonds, and 695 and 750 cm^−1^ bands demonstrate monosubstituted phenyl ring vibrations. The 900–1120 cm^−1^ band shows the P-O stretching vibration of the tetrahedral C-PO_3_. In addition, 1165 cm^−1^ signifies the existence of P-C stretching vibration. There is no doubt that the bands of 1485 and 1435 cm^−1^ are skeleton vibrations of the aromatic ring. Furthermore, a common wide stretched area of 3200–3600 cm^−1^ shows that they all have -OH species. The band has a blue shift from 520 cm^−1^ of HfO_2_ to 560 cm^−1^ of PhP-Hf. This reveals the connection between Hf-O and P species [40].

Figure 2b shows the XRD pattern, and it can be concluded that HfO_2_ is crystalline with tetragonal (t) and monoclinic (m) structures. On the contrary, PhP-Hf (1:1.5) has an amorphous structure. It is worth noting that, in its diffraction pattern, an additional reflection with a d-spacing of 14.9 Å was observed at 2θ of 5.9°. It is probably assigned to the distance between Hf layers separated by PhP [41].

The catalyst pore structure characteristics can be clarified by N_2_ adsorption–desorption isotherms. As shown in Figure 2c and Table 1, HfO_2_ features a curve H1-type loop and shows the D_mean_ (average pore size) of 28.8 nm that is smaller than that of PhP-Hf (1:1.5) (3.5 nm). It is indicated that PhP-Hf (1:1.5) with an H4-type loop is mesoporous, playing a positive role in the adsorption, separation, and catalytic reaction of macromolecules that are difficult to contact with many microporous materials. PhP-Hf (1:1.5) has a higher S_BET_ (BET surface area) and V_pore_ (volume of pores) (213 m^2^/g, 0.22 cm^3^/g) than those of HfO_2_ (24 m^2^/g, 0.16 cm^3^/g). These results confirm that PhP-Hf (1:1.5) forms an additional intermediate layer through assembly, which greatly improves its surface texture, with more active sites available to the substrate.

Through XPS quantitative analysis, it is found that, in the PhP-Hf (1:1.5), the molar ratio of P/Hf is 1.07 (epiphase), which is much higher than that obtained by ICP analysis (0.68, bulk phase). This difference indicated that Hf was mainly encapsulated in PhP-Hf (1:1.5). Conversely, most of the organic ligands (PhP) are located on the hybrid surface.

TG analysis was performed with the programmed temperature range of 50–600 °C to analyze the thermal stability of two catalysts. Figure 2d shows that catalysts have almost no weight loss up to 300 °C, which proves their good thermal stability. In a relatively high-temperature range (300 to 550 °C), the weight begins to lose, which is due to the removal of weak bond -OH species after the temperature rises. Until the temperature rises above 550 °C, the organic matter begins to decompose, resulting in a greater weight loss of PhP-Hf (1:1.5).

Nano-sized materials and structures of catalysts were examined by HR-TEM (Figure 3), showing amorphous structure. This result is in agreement with the crystal structure measured by XRD (Figure 2b). The worm-like stripes shown in Figure 3A can be a result of the disordered interlayers [42]. The particle size is estimated to be ca. 14 nm, indicating the nanostructure of PhP-Hf (1:1.5). The STEM-HAADF diagrams and the elemental mappings in Figure 3 show that P, O, Hf, and C elements are uniformly dispersed on the surface of the catalyst PhP-Hf (1:1.5), which is well connected to the entire nano-hybrid material. This also further proves that the catalyst preparation is successful.

As shown in the XPS spectra (Figure 4a), the binding energy of Hf 4f in PhP-Hf (1:1.5) (17.2 and 19.0 eV) is slightly larger than that in HfO_2_ (17.0 and 18.5 eV) respectively, which indicates the more positive charge of Hf species, correlated with the formation of Hf with relatively stronger Lewis acidity. For the O 1s orbital (Figure 4b), the binding energy (530.5 eV) of the P-O-Hf interaction is relatively higher than that (531.2 eV) of the Hf(H)-O-Hf combination mode. This is an account for a smaller negative charge on the oxygen element. In other words, PhP-Hf (1:1.5) has a lower base strength than HfO_2_. Thus, the introduction of organic ligand PhP is a good way to increase the acid content and strength of PhP-Hf (1:1.5). In addition, the formed P-O-Hf framework with interlayers also increases the overall surface area and pore volume of PhP-Hf [43]. The relatively stronger basicity of the industrial catalyst HfO_2_ may be derived from the Hf-O-Hf framework or hydroxide.

From TG analysis (Figure 2c), it can be found that the PhP-Hf is relatively stable below 300 °C. Therefore, CO_2_-TPD (Figure 5a) and NH_3_-TPD (Figure 5b) are conducted in the programmed temperature range of 50 to 300 °C. The acidic and alkaline densities were evaluated accordingly. Furthermore, the obtained results are shown in Table 2. Both acidic and basic densities of PhP-Hf (1:1.5) (0.27 and 0.32 mmol/g) are higher than those of HfO_2_ (0.16 and 0.24 mmol/g). Even after repeated use of PhP-Hf (1:1.5), basicity increased from 0.32 to 0.35 mmol/g. On the other hand, the acidic sites are reduced from 0.27 to 0.25 mmol/g, and only slight changes took place between the two. It is noteworthy that the abundant acidic site content may cause the Hf-OH and P-OH groups to be converted into additional acidic species. TOF values and selectivity of the catalysts in different P/Hf ratios have been comprehensively considered, demonstrating that the catalyst with a P/Hf ratio of 1:1.5 (i.e., PhP-Hf (1:1.5)) is the best option.

### 3.2. Activity of Different Catalysts

Compared with using H_2_ as a hydrogen source in the autoclave, the advantage of transfer hydrogenation is that the reaction is relatively safer under milder pressures. At the same time, the equipment requirements are not high, and the hydrogen source used is cheap and easy to obtain. Hereby, we screened the hydrogen supply sources, including methanol, ethanol, 1-propanol, 2-propanol, and 2-pentanol, which were also used as solvents to provide hydrogen protons under optimal reaction conditions (Figure 6). Although other alcohols can also impart hydride for the reduction of 4-MOPP, the obtained AN selectivity is not comparable to that of 2-pentanol. The choice of solvent is that secondary alcohols are based on the fact that secondary alcohols are more effective than primary alcohols in the CTH reaction. Gratifyingly, 2-pentanol proved to be the best hydrogen donor, which can be ascribed to its relatively lower reduction potential (67.9 kJ mol^−1^) and high boiling point (118–119 °C).

The amount of catalyst is also an important indicator of its activity to the specific reaction. Figure 7 reveals the influence of catalyst dosage under optimal reaction conditions (i.e., 1 mmol 4-MOPP, 10 mL 2-pentanol, PhP-Hf (1:1.5) catalyst, 220 °C for 2 h.). Surprisingly, a very small amount of catalyst (0.025 g) can achieve a relatively considerable yield (64.7%). With every increase of 0.025 g catalyst, ether is further converted to AN, and its yield reaches the highest value (98.1%) at 0.1 g. These results all indicate that the catalyst has an irreplaceable effect on promoting the conversion of 4-MOPP. However, after the amount of catalyst is more than 0.1 g, a further increase in the yield of AN is not observed. In addition, an optimal TOF value of 8.5 h^−1^ could be obtained. This shows that the use of PhP-Hf (1:1.5) catalyst can increase the reaction rate of the MPV reduction step.

### 3.3. Effect of Reaction Temperature and Time

Figure 8 demonstrates the effect of reaction temperature (120–220 °C) and time (1–3 h) on the reaction process. Obviously, as the temperature increased, the selectivity and yield of *cis*-AN and *tans*-AN continued to increase, as did the conversion of 4-MOPP. It can be seen that ether was the dominant byproduct (up to 30% yield) especially in the initial stage of reaction (0–2 h) at a low temperature of 150–190 °C. Brønsted acid or Lewis base sites play a significant role in this duration. In addition, a phenomenon attracted our attention, where ether with increasing reaction time and temperature could be further converted into AN. AN was formed in high yields of up to 98.1% with 99.8% 4-MOPP conversion was obtained after reaction for 2 h at 220 °C, which was possibly attributed to the promotion of the reduction reaction by the Lewis acid sites. It is worth noting that all the experiments performed detected a mass balance of nearly 100%. From the above data, it can be inferred that PhP-Hf (1:1.5) is a bifunctional and recyclable Hf-based nanocatalyst, which has a significant catalytic effect in tandem MPV reduction and dehydration reactions, showing its potential application prospects in the efficient thermo-conversion of biomass feedstocks.

### 3.4. Catalyst Recycle Study

Whether a catalyst can be reused is an important indicator to evaluate its heterogeneity and practical implications. To prove the heterogeneity and recyclability of the PhP-Hf (1:1.5) catalyst, two sets of parallel experiments were designed. After stirring for 1 h under the above-optimized conditions, the PhP-Hf (1:1.5) catalyst was hot-filtered out in group (a), while, in group (b), the reaction continued to react for another 1 h. Surprisingly, it continued to catalyze the conversion of 4-MOPP to AN in group (b), while almost no reaction was observed in group (a). Moreover, almost no Hf species in the remaining reaction solution was detected by ICP analysis, showing that it has good heterogeneity. After each cycle of the reaction, the solid catalyst residue in the solution was separated by centrifugation. Then, the catalyst was successively washing with DMF, ethanol, and methanol under ultrasonic treatment. The recycled PhP-Hf (1:1.5) was directly used for the next run after being dried at 80 °C in a drying oven (DZF-6020). It can be clearly seen in Figure 9 that the conversion of 4-MOPP and the yield and selectivity of AN do not have a notable decrease after five repeated runs. This further proves the good stability of the PhP-Hf (1:1.5).

To test the change of catalyst structure before and after the reaction, PhP-Hf (1:1.5) recovered after reusing five cycles was characterized by FT-IR (Figure 10a), TG-DTG (Figure 10b), XRD (Figure 10c), and XPS (Figure 10d,e). Figure 10a shows that the catalyst can still maintain its complete structure after being recycled five times. Figure 10b shows that the recovered catalyst has a small amount of loss after 100 °C, while the fresh one starts to lose after 300 °C, indicating that the recycling has a certain impact on the thermal stability. Figure 10c shows the binding energy of Hf 4f increased from (17.38 and 18.88 eV) to (17.78 and 19.38 eV) in recovered PhP-Hf (1:1.5), indicating that the electron cloud density around O decreased, and the corresponding Lewis acidity increased. For Figure 10d, the binding energy of the recovered PhP-Hf (1:1.5) (531.68 eV) is relatively higher than that of the fresh PhP-Hf (1:1.5) (531.28 eV), possibly because of the smaller negative charge on the oxygen species with a lower base strength. It can be seen from Figure 10e illustrates that there is almost no change in crystal form after repeated use. Overall, these figures all prove that there is almost no change after reusing the catalyst, which is a good way to reduce waste and conform to the concept of environmentally friendly chemistry. Therefore, PhP-Hf (1:1.5) can be used as a green bifunctional catalyst for promoting CTH coupled with other cascade reactions.

### 3.5. Reaction Mechanism Study

The poisoning experiment was also conducted, and the results showed that benzoic acid slightly affects the yield of *trans*-AN (Figure 11), and hardly affects the yield of *cis*-AN. In sharp contrast, the addition of pyridine significantly reduced the yield of *trans*-AN from 88.2% to 32.8%. Therefore, it can be inferred that the acidic site plays a more important role in the reaction process. The basic site also has a certain effect on the reaction, but it is not as significant as the acidic site. Since the acid sites of the catalyst are passivated and the formation of homeopathic AN is a kinetic reaction, the effect of transfer hydrogenation is affected.

A possible reaction mechanism was explored (Scheme 2). First, 4-MOPP and 2-pentanol undergo MPV reduction with the catalyst. In this process, a typical six-member transition state is formed. The Lewis acidity of Hf^4+^ helps to increase the activity of the carbonyl group. Through transfer hydrogenation, the carbonyl group on the raw material is converted to a hydroxyl group, forming 4-MOPE, while the 2-pentanol is converted to 2-pentanone. 4-MOPE is further dehydrated under the action of Brønsted and Lewis acids to form *trans*-AN. The main by-product in this reaction is ether, also known as anisole, which is formed by the etherification of the secondary alcohol with 4-MOPE in the presence of the acid catalyst [44]. With the extension of reaction temperature and time, the ether could also be converted to the final product *trans*-AN by the removal of 2-pentanol.

In addition, Table 3 shows that PhP-Hf (1:1.5) has a universal catalytic performance in the transfer hydrogenation and dehydration of carbonyl compounds. The products include olefins, furfuryl alcohol, and γ-valerolactone (GVL), and some of them can be widely used as biofuel precursors.

## 4. Conclusions

A novel approach to efficiently and rapidly prepare AN, which is widely used in food, daily necessities, and medicine for fine synthesis, is proposed. The PhP-Hf (1:1.5) hybrid polymeric catalyst was prepared by a facile solvothermal method at 120 °C. It was demonstrated to have a good catalytic performance in the CTH and dehydration of 4-MOPP (99.8% conversion) to AN (98.1% yield) at 220 °C for 2 h. It was found that PhP-Hf (1:1.5) was a nano-scale (ca. 14 nm) mesoporous material (3.5 nm). In addition, the PhP-Hf (1:1.5) catalyst had both acid (0.27 mmol/g) and base (0.32 mmol/g) sites, which can be responsible for its superior catalytic performance. The investigation of the reaction mechanism shows that Lewis acid (Hf^4+^) and base (O^2−^) sites play a synergistic role in the CTH process and the formation of the six-member transition state. Brønsted and Lewis acid species concurrently help dehydration of 4-MOPP to form *trans*-AN. Moreover, characterization methods and experiments proved that the PhP-Hf (1:1.5) catalyst was highly thermal stable and could be recycled five times with no conscious decline in its activity.

## Data Availability

Not applicable.

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
