# Peer review of "Catalytic Stereoselective Conversion of Biomass-Derived 4′-Methoxypropiophenone to Trans-Anethole with a Bifunctional and Recyclable Hf-Based Polymeric Nanocatalyst"

_polymers, 2021, doi:10.3390/polym13162808_

Round 1
Reviewer 1 Report
The article ‘Catalytic stereoselective conversion of biomass-derived 4’methoxypropiophenone to trans-anethole with a bifunctional Hf-based polymeric catalyst’ by Liu et al. submitted to Polymers details experimental and theoretical studies on a Hf-catalyzed reaction.
I have several serious problems with this manuscript, both conceptually and with the presented results as detailed below:
- According to the authors, this manuscript is an invited contribution to a special issue on “MOF-Based Functional Catalytic Materials for Biofuels Production” of polymers. The manuscript itself has nothing to do with polymers, nor with biofuels. The authors claim that the catalyst is a polymeric material but then why do chemists have coined the term ‘metal-organic framework’ for this? Ordinarily, such a manuscript would be better suited to Catalysts (MDPI) or a similar journal.
- The catalyst to substrate ratio is extraordinarily high (100mg catalyst to 160mg substrate). The TOF indicated by the authors (8) means a turnover number (TON) of 16, i.e., 1 mol Hf for 16 mol substrate. Hafnium is not exactly cheap. If the authors consider this to be an achievement, it needs to be better justified.
- DFT identifies a pathway with a barrier of around 10 kcal/mol, indicating very fast reaction but incompatible with the experimental reality of prolonged heating at 200degC.
- TOFs are only given in the text, not for individual experiments.
- The title of the manuscript implies stereoselectivity. There are no chiral atoms in the substrate or product. What is implied here?
In summary, I do not share the authors' confidence that these results are significant enough. From my point of view, it is entirely out of the usual focus of polymers and at least slightly out of focus of the special issue. Moreover, the DFT study appears not to be correct, and the identified species cannot be active ones (the barrier is approx. 25 kcal/mol too low). This either implies that the model is wrong or that there is a pre-equilibrium that the authors have so far ignored. I do not think that this can be fixed in a short timeframe and recommend rejection of the paper.
Author Response
The article ‘Catalytic stereoselective conversion of biomass-derived 4’methoxypropiophenone to trans-anethole with a bifunctional Hf-based polymeric catalyst’ by Liu et al. submitted to Polymers details experimental and theoretical studies on a Hf-catalyzed reaction.
I have several serious problems with this manuscript, both conceptually and with the presented results as detailed below.
Response: Thank you very much for your valuable comments and suggestions. All the raised major issues have been clarified and resolved carefully, and we believe the revised manuscript can meet the journal's standard and requirements. It would be highly appreciated if you could kindly reconsider our manuscript for possible publication.
- According to the authors, this manuscript is an invited contribution to a special issue on “MOF-Based Functional Catalytic Materials for Biofuels Production” of polymers. The manuscript itself has nothing to do with polymers, nor with biofuels. The authors claim that the catalyst is a polymeric material but then why do chemists have coined the term ‘metal-organic framework’ for this? Ordinarily, such a manuscript would be better suited to Catalysts (MDPI) or a similar journal.
Response: Thank you very much for your kind comments.
- Many papers have demonstrated that MOF is a coordination polymer material with good crystal form. Although these are extended crystal structures and not large discrete molecules such as polymers, they were dubbed coordination “polymers”—a term that is still in use today, although we prefer the more descriptive term MOFs, introduced in 1995 and now widely accepted. (Science 2013, 341 (6149), 1230444.) In Scheme 2, we have perfected the configuration of the catalyst polymer to make its characteristics more obvious (Page 11).
- Based on your suggestions, we have added some other bio-based substrates to produce biofuels, as shown in Table 3 (Page 12).
- The catalyst to substrate ratio is extraordinarily high (100mg catalyst to 160mg substrate). The TOF indicated by the authors (8) means a turnover number (TON) of 16, i.e., 1 mol Hf for 16 mol substrate. Hafnium is not exactly cheap. If the authors consider this to be an achievement, it needs to be better justified.
Response: Thank you for your comments.
- Indeed, the price of transition metal hafnium is not low enough, and we have deleted and revised the relevant contents. However, compared to the noble metal ruthenium catalyst used in the same preparation of anethole system, the price of non-noble metal hafnium is lower (Green Chem. 2011, 13 (2), 307-313.) (Page 2).
- Besides, compared to the systems that also use transition metals as catalysts (Zr-beta and Zr-MSU-3), they require a series of cumbersome steps such as high-temperature calcination and dealumination (ChemSusChem 2018, 11 (17), 3007-3017.). Our preparation method is very simple (directly heating after mixing), giving up to 98.1% yield of product with only 2 h of reaction (Page 2), which is quite shorter than previous reports (e.g., more than 23 h).
- In summary, our work can still be said to be an achievement for this reaction system. Thank you again for your valuable suggestions. (Page 2-3)
- DFT identifies a pathway with a barrier of around 10 kcal/mol, indicating very fast reaction but incompatible with the experimental reality of prolonged heating at 200degC.
Response: Thank you for your suggestion. To avoid some misunderstandings, we convert the unit from 10.17 kcal mol-1 to 42.58 kJ mol-1, which is compatible with the experimental reality. Similar energy barriers at comparable temperatures (180degC, 48.9kJ mol-1; 160degC, 30.9 kJ mol-1; 150degC, 35mol-1) can be found in other MPV hydrogenation processes (ACS Sustainable Chemistry & Engineering 2020, 8 (31), 11477-11490.; International Journal of Hydrogen Energy 2013, 38 (17), 7065-7069.; Fuel 2021, 300, 120996.). As for the difference, it may be due to our calculation taking into account the influence of solvent, which played an important role in lowering down the whole calculation (Page 12).
- TOFs are only given in the text, not for individual experiments.
Response: Thank you for your suggestion. TOF values for the catalysts with different ratios of PhP and Hf have been also calculated and added in Table 2 (Page 7).
- The title of the manuscript implies stereoselectivity. There are no chiral atoms in the substrate or product. What is implied here?
Response: Thank you for your suggestion.
In addition to chiral isomerism, stereoselectivity also has Cis-trans isomers. Cis-trans isomers are stereoisomers, that is, pairs of molecules which have the same formula but whose functional groups are in different orientations in three-dimensional space. You can see the detiled explanation about the definition from Wikipedia (https://en.wikipedia.org/wiki/Cis%E2%80%93trans_isomerism).
Since the anethole also has two isomers, cis-anethole and trans-anethole, and only trans-anethole is safe for the human body and is more widely used. Therefore, we need to use a stereoselective catalyst to maximize the proportion of trans-anethole (Page 2). Thank you.
- In summary, I do not share the authors' confidence that these results are significant enough. From my point of view, it is entirely out of the usual focus of polymers and at least slightly out of focus of the special issue. Moreover, the DFT study appears not to be correct, and the identified species cannot be active ones (the barrier is approx. 25 kcal/mol too low). This either implies that the model is wrong or that there is a pre-equilibrium that the authors have so far ignored. I do not think that this can be fixed in a short timeframe and recommend rejection of the paper.
Response: Thank you very much for your valuable comments and suggestions. All the raised major issues have been clarified and resolved carefully (see details above and here).
- First of all, MOF is a type of polymer, and we have added some examples of catalysts to convert other substrates into biofuels, which is in line with the theme of this special issue. (Page 3)
- We think this may be a unit conversion misunderstanding, and we have converted 10.17 kcal mol-1 to 42.58 kJ mol-1. Different solvents will affect the energy barrier, which may be the reason for the difference from your conclusion. In addition, we have found similar energy barriers (180degC, 48.9kJ mol-1; 160degC, 30.9 kJ mol-1; 150degC, 35mol-1) in the hydrogenation step of transfer hydrogenation in other recent work (ACS Sustainable Chemistry & Engineering 2020, 8 (31), 11477-11490.; International Journal of Hydrogen Energy 2013, 38 (17), 7065-7069.; Fuel 2021, 300, 120996.).
Reviewer 2 Report
In this contribution, authors propose a catalytic study on the valorization of a lignin derived molecule in high-added value product by using the Hf-based polymeric nanocatalyst. Authors provide a careful experimental description supported by a good catalyst characterization and catalytic tests. Results are interesting. The topic is of high relevance, being timely and a hot topic at the moment. Therefore, the manuscript may be of interest for publication in Polymers journal. Therefore, the following points need to be addressed before the publication:
- Introduction section lacks of key references on valorization of lignin derived molecules under CTH condition (see some recent papers published on this topic: ACS Sustainable Chemistry and Engineering, 2021, 9(9), pp. 3379-3407; Catalysis Today, 2020, 357, pp. 511–517; Green Chemistry, 2019, 21(20), pp. 5556-5564).
- All Figures are very small and so difficult to visualize. Authors should improve their quality.
- Some English polishing to correct typos and mistakes is needed.
- Some references must get the right formatting.
Author Response
In this contribution, authors propose a catalytic study on the valorization of a lignin derived molecule in high-added value product by using the Hf-based polymeric nanocatalyst. Authors provide a careful experimental description supported by a good catalyst characterization and catalytic tests. Results are interesting. The topic is of high relevance, being timely and a hot topic at the moment. Therefore, the manuscript may be of interest for publication in Polymers journal. Therefore, the following points need to be addressed before the publication:
Response: Thank you very much for your valuable comments and suggestions. All the below raised major issues have been clarified and resolved carefully.
Namely:
- Introduction section lacks of key references on valorization of lignin derived molecules under CTH condition (see some recent papers published on this topic: ACS Sustainable Chemistry and Engineering, 2021, 9(9), pp. 3379-3407; Catalysis Today, 2020, 357, pp. 511–517; Green Chemistry, 2019, 21(20), pp. 5556-5564);
Response: Thank you for your thoughtful reference to the literature, we have cited and discussed them in the introduction (Page 3, Refs 33-35).
- All Figures are very small and so difficult to visualize. Authors should improve their quality;
Response: Thanks for your suggestion. All the figures have been replaced with the high-quality and high-resolution images.
- Some English polishing to correct typos and mistakes is needed;
Response: We have carefully checked and modified the language errors in the manuscript. Thanks.
- Some references must get the right formatting;
Response: We have checked carefully and adjusted the references to the correct format. Thanks.
Reviewer 3 Report
In this article the one-pot preparation of a bifunctional Hf-based polymeric catalyst, by solvothermal method, for the biomass-derived 4’-methoxypropiophenone conversion to anethole, via cascade Meerwein-Pondorf-Verley reduction and dehydration reactions, has been reported. The results of DFT calculations were also reported, and a possible reaction pathway was proposed.
In general I think this work is interesting and the results are consequential to the good research design. All aspects of the research are well developed, however I suggest adding a comparison with other methods of anethole preparation. Something has already been written in the introduction, however a rough economic evaluation, for example on raw materials or perhaps on the process, would be extremely useful to understand if this method is actually competitive.
Correlation cards used for assigning XRDs should be added.
Line 185: The TG proves a good stability? Which kind of stability?
Figure 3: The resolution of mapping is low. It would be nice to have better images.
The conclusions are poor, all the main results of both characterization and catalytic activity should be reported
Author Response
In this article the one-pot preparation of a bifunctional Hf-based polymeric catalyst, by solvothermal method, for the biomass-derived 4’-methoxypropiophenone conversion to anethole, via cascade Meerwein-Pondorf-Verley reduction and dehydration reactions, has been reported. The results of DFT calculations were also reported, and a possible reaction pathway was proposed.
Response: Thank you very much for your valuable comments and suggestions. All the raised major issues have been clarified and resolved carefully.
Namely:
- In general I think this work is interesting and the results are consequential to the good research design. All aspects of the research are well developed, however I suggest adding a comparison with other methods of anethole preparation. Something has already been written in the introduction, however a rough economic evaluation, for example on raw materials or perhaps on the process, would be extremely useful to understand if this method is actually competitive.
Response: Thank you very much for your valuable comments and suggestions. All the raised major issues have been clarified and resolved carefully. (Page 2)
- Based on your suggestion, we have added a comparison of other methods of preparing anethole. They are the traditional natural distillation method, the precious metal ruthenium catalyst system (Green Chem. 2011, 13 (2), 307-313.), and the Zr-beta zeolite catalyst system (ChemSusChem 2018, 11 (17), 3007-3017.). Our advantages are explained in terms of yield, economic rationality, and simplicity of the preparation method.
- According to the statistics from the General Administration of Customs of China (GACC), China’s imports and exports of unground star anise in 2020 are about 58 million dollars in total, so the demand has potential and prospects for economic application.
- Correlation cards used for assigning XRDs should be added.
Response: We appreciate your valuable comments and suggestions. The correlation cards used for assigning XRDs already be added (Page 5).
- Line 185: The TG proves a good stability? Which kind of stability?
Response: Thank you for your very constructive suggestions. TG thermogravimetric analysis shows that its thermal stability is good. There is almost no loss at 300°C, and the mass loss starting at 300-500°C can be attributed to the decomposition of organic matter. The relevant contents have been added in the revised manuscript (Page 5).
- Figure 3: The resolution of mapping is low. It would be nice to have better images.
Response: Thank you for your very constructive suggestions. The images with better resolution have been provided. (Page 6).
- The conclusions are poor, all the main results of both characterization and catalytic activity should be reported.
Response: Thank you for your very constructive suggestions. We have added the relevant results of characterization and catalyst activity in the conclusion section (Page 13).
Round 2
Reviewer 1 Report
I have indicated in the previous round that I recommend rejection of the manuscript and submission to a different journal. Two other reviewers disagree with my assessment, so I will leave it at that and respect their decisions.
The authors have now provided a revised version. My comment regarding the DFT study and the calculated barrier height was addressed converting kcal/mol to kJ/mol with the authors claiming that the conversion was the likely source of confusion. Let me be very clear here. It was not. The authors claim that the barrier for the rate limiting step is around 10 kcal/mol (42 kJ/mol). The lowest know barriers for the fastest polymerization catalysts are 10-12 kcal/mol, below that, rates become diffusion controlled. Even at -100degC rates for such a reaction are too high to trap intermediates. The authors have to heat there system to 220deg for the reaction to occur in 2h. One can estimate that the barrier for the rate limiting step MUST be around 50 kcal/mol then (10.1021/acs.organomet.8b00456). The authors give a reference to back up their claims where an apparent activation barrier of similar magnitude is reported. However, that is Ea (from the Arrhenius equation) and A (the prefactor) remains unknown. In the present paper the authors give a Gibbs free energy barrier which is far too low. The paper should not be published without this problem being addressed. I suspect that the authors are missing a pre-equilibrium which would factor into the overall barrier.
Author Response
Thank you very much for your valuable comments and suggestions. As you kindly pointed out, DFT simulated by Gaussian software may have some unknown deviations. To avoid misleading readers, relevant data and discussions have been deleted in the revised manuscript. Thank you for your strict control of the quality of the article.
Reviewer 2 Report
Authors have revised the manuscript thoroughly, according to reviewers' comments and suggestions. I recommend this paper for publication in Polymers journal.
Author Response
Thank you very much for your comments and kind recommendation.
Round 3
Reviewer 1 Report
The authors have addressed my concerns by removing the DFT part. In my opinion, the manuscript can now be accepted.